# Recent Progress in Innate Immune Responses to Enterovirus A71 and Viral Evasion Strategies

**DOI:** 10.3390/ijms25115688

**Published:** 2024-05-23

**Authors:** Jialong Wei, Linxi Lv, Tian Wang, Wei Gu, Yang Luo, Hui Feng

**Affiliations:** 1School of Medicine, Chongqing University, Chongqing 400044, China; 202337021003@stu.cqu.edu.cn (J.W.); l18523921569@163.com (L.L.); wangtian211002@163.com (T.W.); guwei@cqu.edu.cn (W.G.); 2Institute of Precision Medicine, Chongqing University, Chongqing 400044, China

**Keywords:** enterovirus, innate immune response, hand, foot, and mouth disease (HFMD), evasion strategies

## Abstract

Enterovirus A71 (EV-A71) is a major pathogen causing hand, foot, and mouth disease (HFMD) in children worldwide. It can lead to severe gastrointestinal, pulmonary, and neurological complications. The innate immune system, which rapidly detects pathogens via pathogen-associated molecular patterns or pathogen-encoded effectors, serves as the first defensive line against EV-A71 infection. Concurrently, the virus has developed various sophisticated strategies to evade host antiviral responses and establish productive infection. Thus, the virus–host interactions and conflicts, as well as the ability to govern biological events at this first line of defense, contribute significantly to the pathogenesis and outcomes of EV-A71 infection. In this review, we update recent progress on host innate immune responses to EV-A71 infection. In addition, we discuss the underlying strategies employed by EV-A71 to escape host innate immune responses. A better understanding of the interplay between EV-A71 and host innate immunity may unravel potential antiviral targets, as well as strategies that can improve patient outcomes.

## 1. Introduction

Enterovirus A71 (EV-A71) is a non-enveloped virus that belongs to the genus *Enterovirus* of the family Picornaviridae. It was initially isolated from a child with a disease of the central nervous system in California in 1969 [1]. As a major pathogen causing hand, foot, and mouth disease (HFMD), EV-A71 primarily affects infants and young children under the age of five [2]. With the eradication of wild poliovirus (the most well-known and best-studied enterovirus), EV-A71-associated HFMD has emerged as a serious public health concern, particularly in the Asia–Pacific region [3,4]. Although EV-A71 infections are generally mild and self-limiting, severe and life-threatening neurological complications, such as aseptic meningitis, cerebellar ataxia, acute flaccid paralysis, encephalitis, and pulmonary edema, have been reported frequently [5,6,7]. More recently, EV-A71 was also identified in patients in the United States with confirmed acute flaccid myelitis (AFM), a disabling poliomyelitis-like illness most often linked to EV-D68 infections [3,8]. In fact, EV-A71 is currently considered the most neurotrophic non-polio enterovirus [9]. Strikingly, despite many efforts, only three inactivated EV-A71 vaccines have been licensed exclusively by China’s National Medical Products Administration (NMPA), and clinically approved direct antiviral agents/treatments are still lacking [10].

The pathogenesis of EV-A71 infection is largely unknown at present. Generally, host immunity and viral factors, as well as their interactions, influence the pathogenesis and outcomes of virus infection. The innate immune system stands as the first defense line to detect and respond to viral infections. Accordingly, there has been a substantial amount of research unveiling the mechanisms by which EV-A71 infections induce host innate immune responses [11]. In particular, while our understanding of the pathogen-triggered immunity (PTI; e.g., viral RNA) has significantly improved over the past decade, there is also a growing appreciation of the impacts of effector-triggered immunity (ETI) that senses the activity of pathogen-encoded effectors [12]. In parallel, it is well known that EV-A71 has developed sophisticated strategies to evade antiviral innate immunity. For instance, it encodes two viral proteases that antagonize host innate immune signaling, as described in detail below. Here, we review the past epidemiology of EV-A71-associated HFMD, the molecular virology and life cycle of EV-A71, and recent progress in understanding how EV-A71 infections trigger host innate immune responses. Moreover, we discuss how EV-A71 targets the innate immune signaling pathways as a counteraction to favor viral replication and propagation.

## 2. Past Epidemiology of EV-A71-Associated HFMD

Since its discovery in 1969, there have been frequent large-scale outbreaks of HFMD caused by EV-A71 infections in various regions of the world. For instance, outbreaks of EV-A71 infection have been documented in the United States and European countries, such as Sweden, Bulgaria, Hungary, and the Netherlands [13,14,15,16,17]. However, it is important to note that HFMD and its characteristic symptoms were officially recognized as early as 1957 [18,19]. Meanwhile, poliovirus was once one of the most devastating public health problems [20]. Thus, the health burden caused by EV-A71 was likely greatly underestimated.

Since the late 2000s, the Asia–Pacific regions have suffered from sporadic HFMD outbreaks linked to EV-A71 infections, with regional epidemics recurring approximately every three years [21]. For example, a severe outbreak of EV-A71-associated HFMD occurred in Taiwan in 1998, resulting in 78 deaths among 405 children with neurological complications. It escalated into a larger outbreak in 2000, with 41 deaths among 80,677 cases of HFMD [22,23]. Likewise, there have been repeated outbreaks of HFMD in various regions of China from 2008 to 2014, with a total of over 10 million cases and 3046 deaths [4]. More recently, sporadic cases of EV-A71 infections have also been reported in European countries, including Denmark, France, Germany, Spain, and Poland [24,25,26,27,28]. For instance, the national enterovirus surveillance system in Denmark tested a total of 1143 enterovirus-positive individuals from 1 January 2009 to 31 December 2013, and 63 of them were related to EV-A71 infection [24]. In France, eight cases of EV-A71 infection were diagnosed in Marseille over the period of 2019 to 2020 [25]. However, it should be noted that these numbers only represent confirmative cases in certain regions. As mentioned earlier, it is reasonable to assume that the incidence of EV-A71-related HFMD may be underestimated in these countries. In summary, the past decades have witnessed a significant increase in the pandemic activity of EV-A71 throughout the world.

## 3. The Molecular Virology and Life Cycle of EV-A71

Decades of intensive research on poliovirus, as well as other picornaviruses, have greatly advanced our understanding of the molecular characteristics and replication cycle of non-polio enteroviruses. The virion of EV-A71 consists of an icosahedral capsid shell surrounding a single strand of a positive-sense RNA genome, with a diameter ranging from 24 to 30 nm. Its capsid is composed of 60 copies of four structural proteins (VP1-VP4) arranged with a pseudo T = 3 symmetry, an architecture shared by all enteroviruses [2]. The EV-A71 genome is approximately 7.4 kilobases (kb) in length, and like all picornaviruses, it consists of an internal ribosome entry site (IRES)-containing 5′ untranslated region (UTR), a large open reading frame (ORF) encoding a single polyprotein precursor (Figure 1), and a 3′ UTR with a poly-A tail of varying lengths [29].

EV-A71’s life cycle begins with cell surface receptors’ binding-dependent endocytosis (Figure 1①). It has been shown that scavenger receptor B2 (SCARB2) and P-selectin glycoprotein ligand-1 (PSGL-1) are the primary receptors of EV-A71 [30,31]. Notably, both receptors are expressed in human neuron and glial cells [32], although it is uncertain whether this accounts for the neurological complications caused by EV-A71 infection. Once the virus enters a host cell, it releases its RNA genome into the cytoplasm through a pore on the endosomal membrane [33,34,35] (Figure 1②). This RNA genome is then directed to the polyprotein translation and co-translationally proteolytic cleavage process, generating all mature viral proteins and functional cleavage intermediates required for further replication (Figure 1③). Like all positive-strand RNA viruses, this occurs in close association with replication organelles (infection-induced membrane structures) and in a non-conservative manner (i.e., new positive strands outnumbering new negative-strand RNA, as shown in Figure 1④). The newly synthesized RNAs are then packaged into new virions before exiting the host cell (Figure 1⑤). Intriguingly, although enteroviruses have been recognized as non-enveloped lytic viruses released from ruptured host cells, accumulating evidence suggests that they can also exit infected cells nonlytically in extracellular vesicle structures [36,37,38] (Figure 1⑥).

## 4. Innate Immune Responses to Enterovirus Infections

Enteroviruses are transmitted either via the fecal–oral route or through respiratory droplets, encountering the epithelial cells lining the mucosal surfaces of the gastrointestinal and respiratory tracts [9,39]. Accordingly, the mucosal immune system of these sites plays a crucial role in host defense against enteroviruses. Notably, while HFMD cases are more frequent in older children, the highest susceptible rates occur in children under 5 years old that are known for their immature immune system and for clustering at the preschool level [40,41]. Likewise, many enterovirus infections go unnoticed and are cleared before the production of antibodies [42], indicating that the host innate immunity comprises the primary protection from enteroviruses (summarized in Figure 2).

During infection, epithelial cells geared towards rapid responses detect pathogen-associated molecular patterns (PAMPs) via pattern recognition receptors (PRRs). Like all other positive-stranded RNA viruses, enteroviruses are mainly recognized by the membrane-associated Toll-like receptors (TLRs) and the cytoplasmic retinoic-acid-activated gene I (RIG-I)-like receptors (RLRs) [43,44,45]. Specifically, TLRs and RLRs recognize PAMPs including the viral RNA genome released in host cells and the double-stranded RNA (dsRNA) generated during viral replication (Figure 2). Upon RNA binding, TLRs recruit myeloid differentiation primary-response protein 88 (MyD88) or Toll/interleukin (IL)-1 receptor domain-containing adaptor-protein-inducing interferon-β (TRIF), and RLRs recruit the mitochondrial antiviral signaling protein (MAVS). This then initiates downstream signaling cascades involving multiple kinases that activate nuclear factor kappa-B (NF-κB) and IFN regulatory factor 3 (IRF3), leading to the production of interferons (IFNs) and inflammatory cytokines (Figure 2). Our recent studies have also shown that IRF1 contributes non-redundantly to an immediate IFN response downstream of NF-κB [46,47]. Regarding EV-A71, it has been demonstrated that TLRs, including TLR3 [48], TLR7 [49], and TLR9 [50], as well as the RLR protein, namely melanoma differentiation-associated gene 5 (MDA5) [51,52], all mediate IFN responses against the virus (Figure 2a,b). Notably, RIG-I (a classic RLR sensor) seems to play a small role in this process [52]. This likely reflects distinct RNA binding preferences of RIG-I and MDA5, as the latter has a higher recognition ability for dsRNA longer than 2 kb [53,54,55]. Besides RLRs and TLRs, a recent study also suggested that EV-A71 infection can activate STING (cyclic GMP-AMP receptor stimulator of IFN genes) in a cGAS (cyclic GMP-AMP synthase)-dependent manner (Figure 2b) [56]. Nonetheless, it should be noted that this is due to infection-induced mitochondrial damage and the discharge of mitochondrial DNA into the cytosol of infected cells.

Apart from TLR- and RLR-initiated IFN responses, another PRR family, comprising nucleotide-binding oligomerization domain (NOD)-like receptors (NLRs), is widely recognized for driving inflammatory responses during virus infection [57,58]. Among them, NLRP3 (NLR family pyrin domain-containing 3) has been shown to be activated by enteroviruses via the recognition of virus replication (i.e., dsRNA) and self-derived stress signals, resulting in the secretion of cytokines and inflammatory cell death [59,60,61] (Figure 2c). Indeed, NLRP3 inflammasome was shown to play a protective role against EV-A71 infection in a mouse model, and its activation is triggered by virus replication [61]. Apart from the best-studied NLRP3, a recent study demonstrated that human NLRP1 is a direct dsRNA sensor, triggering the formation of corresponding inflammasome and thereby activating the production of proinflammatory cytokine IL-1β [62] (Figure 2c). However, it is important to note that some NLRs also play crucial roles in regulating innate antiviral immune responses [46,62,63].

In addition to the extensively studied innate immunity through the detection of highly conserved PAMPs, there is growing evidence of the role of ETI that senses the activity of pathogen-encoded effectors [12,64,65]. Importantly, inflammasome-assembling PRRs, CARD8 (caspase recruitment domain family member 8), and NLRP1 have emerged as innate immune sensors of the enzymatic activities of diverse viral proteases [66,67]. For instance, a recent study showed that NLRP1 is activated by the human rhinovirus (an enterovirus causing common cold) [68]. Mechanistically, the viral 3C protease (3C^pro^) specifically cleaves NLRP1 at the Q130^G131 site, a sequence that mimics the preferred cleavage motif shared by multiple enteroviruses [69,70]. Unlike other host proteins being targeted by viral proteases, this reveals that G131 is recognized by the N-terminal glycine degron pathway and subsequent proteasome-dependent degradation, thereby leading to immune activation and cell death [68,71]. Not surprisingly, EV-A71 3C^pro^ similarly cleaves human NLRP1 and triggers the formation of NLRP1 inflammasome, albeit not as robust as that seen for human rhinovirus [70]. Likewise, the CARD8 inflammasome can be activated through sensing the 3C^pro^ from coxsackie virus B3 [72]. Although this reaction has not been verified for EV-A71, a recent study identified a single-nucleotide polymorphism in human CARD8 that enables the sensing of rhinovirus 3C^pro^ activity [73].

IFNs are crucial in host antiviral immunity. They trigger Janus kinase (JAK)- and signal transducer and activator of transcription (STAT)-dependent signaling cascades after binding to their respective receptors, leading to the transcriptional activation of hundreds of IFN-stimulated genes (ISGs, Figure 2d). Importantly, although most cells produce both type I and type III IFNs, there is growing evidence that type III IFNs mediate quicker antiviral responses than type I IFNs at epithelial barriers [74,75,76]. Indeed, recent studies show that EV-A71 predominantly triggers type III IFNs in human intestine epithelial cells, primary human intestinal epithelial monolayers, as well as human and mouse intestine tissues [48,77]. Given that EV-D68 induces a robust type of III IFN response at the respiratory epithelium, this may also occur at the respiratory tracts [78].

## 5. Innate Immune Evasion Strategies of EV-A71

As described above, the innate immune system at the mucosal surfaces of the gastrointestinal and respiratory tracts provides the first line of host defense against enterovirus infections. Not surprisingly, viruses must cross this barrier before they can establish productive infection and spread to other tissues and organs. Indeed, EV-A71 has developed sophisticated strategies that allow it to evade innate antiviral immune responses and facilitate viral propagation. A thorough understanding of these strategies is crucial in order to comprehend the pathogenesis of EV-A71 infection, as well as to develop effective antiviral therapeutics. Hence, we review the diverse viral evasion strategies in this section (Figure 3, Table 1).

### 5.1. Viral Evasion Strategies Based on Manipulating PRRs

Although RNA viruses are primarily detected by RLRs, including RIG-I and MDA5, RIG-I seems dispensable for the detection of EV-A71 [51,52]. Strikingly, there is clear evidence that RIG-I activity is antagonized by EV-A71 [79] (Table 1). For example, the virus encodes a nonstructural protein, namely 2C, which efficiently induces RIG-I degradation through the lysosomal pathway [80]. RIG-I was also shown to undergo caspase- and proteasome-independent proteolytic degradation in EV-A71-infected cells, and this is attributed to a specific cleavage by viral 3C^pro^ [81] (Figure 3). Besides this, the 3C^pro^ is capable of associating with RIG-I via the caspase activation and recruitment domain, precluding the recruitment of its downstream adaptor MAVS [82]. In addition to acting directly on RIG-I itself, EV-A71 has developed ways that indirectly regulate RIG-I function. For instance, it was found that the EV-A71 infection of human rhabdomyosarcoma cells inhibits the production of type I IFNs by downregulating the ubiquitination level of RIG-I [83]. In agreement with this finding, CYLD, a deubiquitinase that removes the K63-linked polyubiquitin chains from target proteins (including RIG-I) and thus inhibits RIG-I-initiated signaling [84], was found to be specifically upregulated by EV-A71 3C^pro^ [85]. EV-A71 also encodes a 2A protease (2A^pro^), and it was shown to cleave DDX6, an RNA helicase that displays anti-EV-A71 activity by reinforcing RIG-I signaling [86] (Figure 3).

Similar to RIG-I, MDA5 was shown to undergo proteolytic degradation in a caspase- and proteasome-independent manner (Figure 3). However, this is mediated by EV-A71 2A^pro^ rather than 3C^pro^ [81]. Notably, the virus also encodes an RNA-dependent RNA polymerase (RdRp; also denoted 3D^pol^), which can interact with the caspase activation and recruitment domains of MDA5, thereby inhibiting MDA5-initiated IFN responses [87].

As for the TLRs, it was shown that EV-A71 can induce autophagy to benefit virus replication in human bronchial epithelial cells. Interestingly, this is accompanied by the degradation of endosomes, leading to a non-specific suppression of the TLR7-mediated type I IFN response [88] (Figure 3). Given that TLR family members, such as TLR3, TLR8, and TLR9, are all present in the endosomal compartments, it is tempting to assume that other endosomal TLRs can also be inhibited by EV-A71 via inducing autophagy. There is also evidence that EV-A71 infection induces the expression of sex-determining region Y-box 4 (Sox4), which in turn suppresses TLR signaling cascades to promote virus replication [89]. Strikingly, this occurs at multiple levels and involves several distinct mechanisms, including the transcription inhibition of TLR genes via the binding of Sox4 to their promoters [89]. Notably, there is currently no evidence that EV-A71-related structural and/or nonstructural proteins directly target TLR sensors. Furthermore, most research in this field has primarily focused on the effects of EV-A71 on downstream adaptor proteins associated with TLR signaling.

Despite the protective role of NLRP3 inflammation in limiting EV-A71 replication, a recent study proposed that the virus may manipulate NLRP3 to promote its own spreading [90]. Intriguingly, this involves virus 3D^pol^ that was found to interact with NLRP3 to facilitate the assembly of the inflammasome complex. Equally remarkable, it was shown that EV-A71 can also counteract inflammasome activation through NLRP3 cleavage at the G493^L494 or Q225^G226 junction as a result of virus 2A^pro^ and 3C^pro^ [61]. Furthermore, EV-A71 3C^pro^ was shown to interact with the NLRP3 protein, leading to the inhibition of IL-1β secretion [61] (Figure 3). In conclusion, EV-A71 seems to have evolved to manipulate NLRP3 inflammasome selectively, and this may have important implications on its pathogenesis (Table 1).

### 5.2. Viral Strategies to Disrupting Key Adaptors of the PRR Signaling Pathways

Following the sensing of PAMPs, the PRR proteins recruit different downstream adaptors to activate IFN responses. Strikingly, these adaptor proteins are also frequently targeted by the virus (Figure 3). For example, MAVS (the key adaptor for cytosolic RNA sensing by RLRs) is cleaved rapidly in EV-A71-infected rhabdomyosarcoma cells [91]. This cleavage occurs independently from cellular apoptosis and proteasome degradation, and is mediated by virus 2A^pro^ [81,92] (Figure 3). Moreover, multiple distinct glysine (G) sites, including G209, G251, and G265 of MAVS, can be targeted, with the highest proteolysis ability at the G251 residue [92]. Similarly, the TLR signaling adaptor TRIF was found to be targeted and cleaved by EV-A71 3C^pro^ at Q312^S313, a proteolytic site shared by the 3C^pro^ of several picornaviruses [71,91,93].

In comparison to adaptor proteins being cleaved directly, MyD88 is targeted by EV-A71 as well, yet it occurs in an indirect manner. For instance, a recent study indicated that virus infection activates Sox4 expression and its binding to the promoter of the *MyD88* gene, resulting in the inhibition of *MyD88* transcription [89] (Figure 3). Furthermore, there is evidence that microRNAs (miRNAs), including miR-30a and miR-21, are regulated by EV-A71 to suppress MyD88 expression [94,95]. However, it should be noted that their routes to exert such a function differ significantly. In particular, miR-30a is preferentially enriched in extracellular vesicles released from virus-infected human oral epithelial cells. Following its delivery into macrophages, the expression of its target genes such as MyD88 is then suppressed [94].

### 5.3. Viral Strategies to Dampen Transcription Factor Activation/Function

NF-κB plays a key role in inflammatory responses to virus infections. Its activation is strictly controlled by the IκB kinase (IKK) complex, which consists of two catalytic subunits (IKKα and IKKβ) and the essential module NEMO (NF-κB essential modulator) or IKKγ [96,97]. Not surprisingly, EV-A71 has adopted multiple strategies to suppress NF-κB activation [98,99,100] (Figure 3, Table 1). It was shown that the 2C protein recruits protein phosphatase 1 to IKKβ, inhibiting its phosphorylation and thus blocking NF-κB activation [98]. Moreover, a recent study reported that, by inducing the formation of inclusion bodies in EV-A71-infected cells, virus 2C protein can also sequester IKKβ and IKKα from phosphorylation through direct interaction with IKKβ [99]. The 2C protein was also reported to reduce the formation of p65/p50 heterodimer (the most predominant form of NF-κB) by interacting with the IPT domain of p65 [101]. In addition to viral nonstructural protein, EV-A71 also takes advantage of host protein to suppress NF-κB activation. In agreement, it was shown that Sox4 expression is induced in virus-infected cells, and its binding to IKKα and IKKβ inhibits their phosphorylation [89].

Similar to NF-κB, the activation of IRF family members such as IRF3 and IRF7 are crucial for IFN responses to virus infection. Likewise, IRFs themselves and/or signaling that leads to their activation have often been identified as targets of EV-A71. For instance, the expression of ubiquitin-specific protease 24 (USP24) was found to be induced by virus infection, and, following physical interaction with TBK1 (a key kinase upstream of IRFs), it can suppress the K63-linked polyubiquitination of TBK1, thereby inhibiting the phosphorylation and nuclear translocation of IRF3 [102] (Figure 3). In contrast, EV-A71 antagonizes IRF7 through the protease activity of 3C^pro^. Mechanistically, the H40 residue at the 3C^pro^ proteolytic active site mediates cleavage at the Q189^S190 junction within the constitutive activation domain of IRF7, sequestering it from activating IFN expression [103].

### 5.4. Viral Strategies to Antagonize IFN-Initiated Immune Responses

As mentioned earlier, IFN responses elicited upon virus infection are crucial for establishing host antiviral innate immunity. Strikingly, although pretreatment with type I IFNs protects mice from EV-A71 infection, it often fails to induce this response [103,104]. Indeed, there is accumulating evidence that the virus applies several different strategies to evade type I IFN-dependent antiviral immune responses (Figure 3). For example, it was suggested that EV7-A71 reduces the IFN receptor 1 (IFNAR1) level through its 2A^pro^ activity [105]. However, this phenomenon has been controversial in human embryonic lung fibroblasts and rhabdomyosarcoma cells, in which the expression of JAK1, but not IFNAR1, is downregulated [106]. Although the mechanisms behind this discrepancy are uncertain, a recent study demonstrated that EV-A71 2A^pro^ can upregulate the expression and secretion of LDL-receptor-related protein-associated protein 1 (LRPAP1), a ligand that triggers IFNAR1 ubiquitination and degradation by binding to its extracellular domain [107] (Figure 3, Table 1).
ijms-25-05688-t001_Table 1Table 1Targeting factors of viral proteins and their strategies to evade innate immunity.Viral ProteinsHost TargetsStrategies to Evade Innate ImmunityReference2A^pro^RIG-ICleaving DDX6 to indirectly inhibit RIG-I signaling[86]NLRP3Cleaving NLRP3 to directly inhibit inflammasome activation[61]MDA5Cleaving MDA5[81]MAVSCleaving MAVS[81,92]IFNAR1Upregulating the level of LRPAP1 to indirectly suppress IFNAR1 expression[105,107] STAT1Reducing the serine phosphorylation of STAT1 to inhibit its nuclear translocation[108]2CRIG-IInducing RIG-I degradation[80]IKKβInhibiting the phosphorylation of IKKβ to block NF-κB activation[98,99]p65Interacting with the IPT domain of p65 to reduce p65/p50 heterodimer formation[101]3C^pro^RIG-ICleaving RIG-I directlyUpregulating the level of CYLD to inhibit RIG-I ubiquitinationAssociating with RIG-I to preclude MAVS recruitment[81,82,85]NLRP1Cleaving NLRP1 directly to inhibit inflammasome activation[69,70]NLRP3Cleaving NLRP3 and interacting with it directly to inhibit inflammasome activation[61]TRIFCleaving TRIF directly[71,91,93]IRF7Cleaving IRF7 directly[103]IRF9Cleaving IRF9 directly[109]ZAPCleaving ZAP directly[110]OAS3Cleaving OAS3 directly[111]3D^pol^MDA5Interacting with MDA5 to inhibit MDA5-initiated IFN responses[87]STAT1Reducing the serine phosphorylation of STAT1 to inhibit its nuclear translocation[108]Abbreviations: DDX6: DEAD-box helicase 6; MDA5: melanoma differentiation-associated gene 5; NLRP1/3: NLR family pyrin domain-containing 3; MAVS: mitochondrial antiviral signaling protein; IFNAR1: type I interferon receptor 1; LRPAP1: LDL-receptor-related protein-associated protein 1; STAT1: signal transducer and activator of transcription 1; RIG-I: retinoic-acid-activated gene I; IKKβ: inhibitor of kappa B kinase β; NF-κB: nuclear factor kappa-B; CYLD: CYLD lysine 63 deubiquitinase; TRIF: Toll/interleukin (IL)-1 receptor domain-containing adaptor-protein-inducing interferon-β; IRF7/9: interferon regulatory factor 7/9; ZAP: zinc-finger antiviral protein; OAS3: 2′-5′-oligoadenylate synthetases 3.


Upon activation, type I IFN receptors signal through IFN-stimulated gene factor 3 (ISGF3), a heterotrimeric complex composed of IRF9 and phosphorylated STAT1 and STAT2. Not surprisingly, IRF9 was shown to be sensitive to EV-A71 3C^pro^ cleavage [109]. Moreover, it was also shown that EV-A71 infection induces a caspase-3-dependent degradation of cellular karyopherin-α1 (KPNA1), a nuclear localization signal receptor for phosphorylated STAT1, which in turn inhibits the transportation of phosphorylated STAT1/2 into the nucleus where they drive the expression of hundreds of ISGs [112]. Similar to type I IFN, type II IFN (IFN-γ) pretreatment reduces EV-A71 propagation. Given the involvement of phosphorylated STAT1 in the IFN-γ signal transduction pathway, it is tempting to anticipate that EV-A71 can also evade type II IFN-dependent innate immunity by downregulating KPNA1. Moreover, both EV-A71 2A^pro^ and 3D^pol^ have been shown to be able to antagonize IFN-γ-induced IRF1 transactivation due to a loss of STAT1 nuclear translocation [108]. Interestingly, while the former accomplishes this by reducing the serine phosphorylation of STAT1 without affecting its expression, the latter is accompanied by a decrease in STAT1 expression [108].

In addition to the above pathways leading to ISG expression, ISG proteins are frequently targeted by EV-A71 to favor virus replication. For instance, the viral 3C^pro^ renders a site-specific proteolysis of two isoforms of host zinc-finger antiviral protein (ZAP), leading to the accumulation of a 40 kDa N-terminal ZAP fragment in virus-infected cells and thus the loss of its antiviral ability [110]. Likewise, 2′-5′-oligoadenylate synthetases 3 (OAS3), an important ISG in the OAS/RNase L antiviral system displaying an obvious inhibitory effect on EV-A71 replication in vitro, was also shown to be targeted by the virus 3C^pro^ for cleavage to enhance viral replication [111]. Moreover, a recent study reported that EV-A71 infection enhances the expression of the suppressor of cytokine signaling (SOCS), a negative feedback regulator of the JAK-STAT signaling, to promote viral infection [113]. Interestingly, this occurs through the NF-κB but not the virus-induced type I IFN pathway.

## 6. Conclusions and Perspectives

In the post-polio eradication era, EV-A71 has emerged as the leading cause of severe HFMD worldwide, and it has been identified in many of the AFM cases lately. Like all RNA viruses, EV-A71 displays high proclivity for the de novo generation of diversity through error-prone genome replication [71,114]. The availability of EV-A71 vaccines in China, but not any other country, increases the probability of pathogen evolution, leading to the emergence of new strains of EV-A71 over time [115]. As such, more research is urgently required to understand the complex molecular interplay between the host and the virus. The virus–host interactions within the innate immune system play a crucial role in the pathogenesis of infectious viral disease. However, we have only begun to understand how EV-A71 induces and antagonizes host innate immunity, and how this is associated with viral pathogenesis in recent years.

Currently, several important questions remain unanswered. For instance, what makes EV-A71 highly neurotrophic? What is the molecular basis behind the phenomenon that EV-A71 infection preferentially triggers type III IFN responses? Does EV-A71 more robustly suppress a type I interferon response than type III IFN response? Can EV-A71 also evade virus protease activity-dependent ETI activation and signal transduction? In addition to the innate immune signal cascades mentioned above, how do other key components of cellular innate immunity contribute to host protection against EV-A71 infection? Answers to these studies will not only help us gain further insights into the innate immune responses to EV-A71, but also facilitate antiviral development to eventually prevent or even eradicate the pathogen.

In conclusion, our knowledge of EV-A71 has increased continually. While outbreaks of EV-A71-related HMFD across the world seem to have declined considerably in recent years, it is important to fully understand the mechanisms of EV-A71 interaction with host innate immunity.

## Figures and Tables

**Figure 1 ijms-25-05688-f001:**
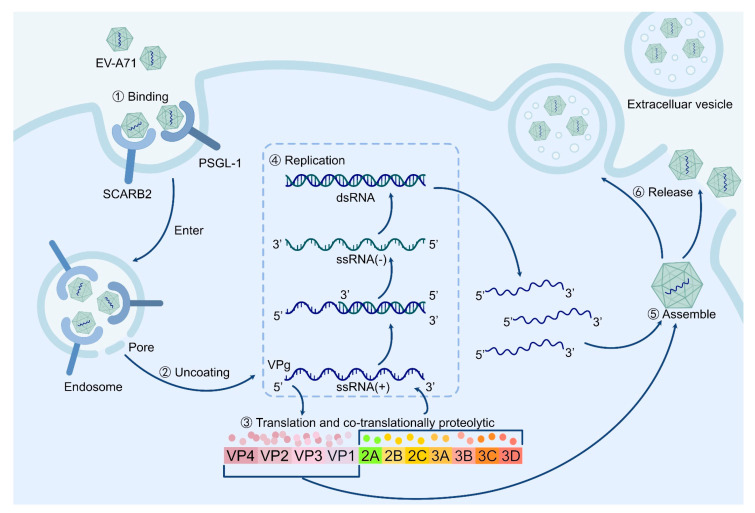
Illustration of the EV-A71 life cycle. EV-A71 enters host cells by binding to receptors such as SCARB2 and PSGL-1, allowing the release of its genomic RNA into the cytoplasm through endosomal membrane pores (①–②). The viral RNA undergoes translation, with VPg covalently linked to it (③). The translated polypeptides are then cleaved into 11 major proteins, including VP1-4 for viral capsid assembly and 2A-3D for the replication of viral RNA genome (④). Finally, the viral RNA and capsid are assembled and processed into mature viruses, which are released through extracellular vesicles or the direct lysis of host cells (⑤–⑥). Abbreviations: EV-A71: enterovirus A71; SCARB2: scavenger receptor B2; PSGL-1: P-selectin glycoprotein ligand-1.

**Figure 2 ijms-25-05688-f002:**
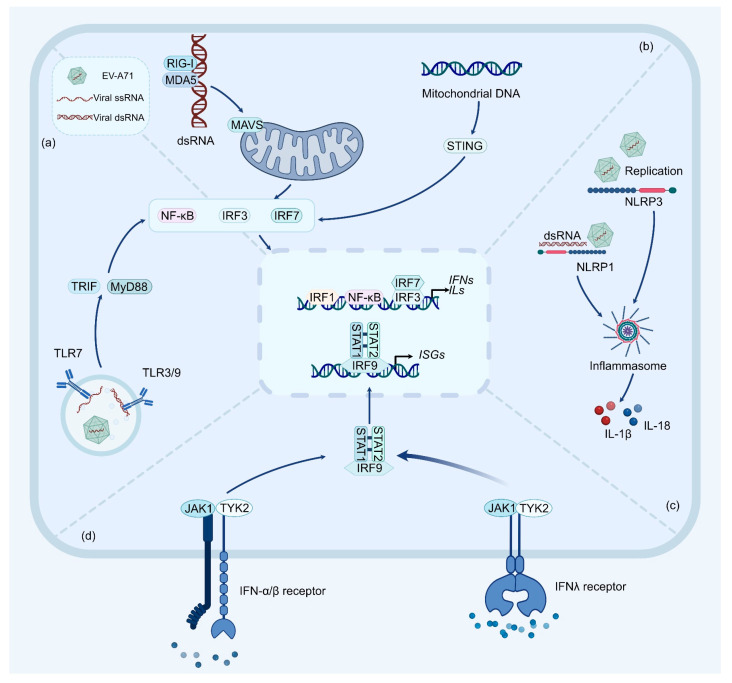
Summary of the innate immune response during enterovirus infection. (**a**) Enterovirus replication activates TLR7 and TLR3/9 through ssRNA and dsRNA, respectively, leading to NF-κB nuclear translocation via TRIF or MyD88 adaptor proteins. (**b**) Cytoplasmic sensors RIG-I and MDA5 are activated by dsRNA, signaling through mitochondrial-associated MAVS and initiating the nuclear translocation of IRF3/7 and NF-κB transcription factors. Enterovirus infection induces mitochondrial damage, releasing mitochondrial DNA into the cytoplasm and activating STING. (**c**) Recognition of virus replication activates NLRP3, forming inflammasomes that activate caspase-1 and secrete IL-1β/IL-18. NLRP1 acts as a sensor for dsRNA and viral protease activity, being activated by enterovirus 3C^pro^ cleavage. (**d**) The binding of IFN-α/β and IFN-λ to their respective IFNARs triggers downstream kinases JAK1 and TYK2, phosphorylating STAT1 and STAT2. This promotes the formation and nuclear translocation of the STAT1-IRF9-STAT2 complex. Abbreviations: ssRNA: single-stranded RNA; dsRNA: double-stranded RNA; TLR3/7/9: Toll-like receptor 3/7/9; RIG-I: retinoic-acid-activated gene I; MDA5: melanoma differentiation-associated gene 5; TRIF: Toll/ interleukin (IL)-1 receptor domain-containing adaptor-protein-inducing interferon-β; MyD88: myeloid differentiation primary-response protein 88; MAVS: mitochondrial antiviral signaling protein; NF-κB: nuclear factor kappa-B; IRF1/3/7/9: interferon regulatory factor 1/3/7/9; STING: stimulator of interferon genes; NLRP1/3: NOD-like receptor (NLR) family pyrin domain-containing 1/3; IL-1β/18: interleukin-1β/18; IFN-α/β: interferon α/β; IFN-λ: interferon λ; IFNARs: type I interferon receptors; JAK1: Janus kinase 1; TYK2: tyrosine kinase 2; STAT1/2: signal transducer and activator of transcription 1/2; ISGs: IFN-stimulated genes.

**Figure 3 ijms-25-05688-f003:**
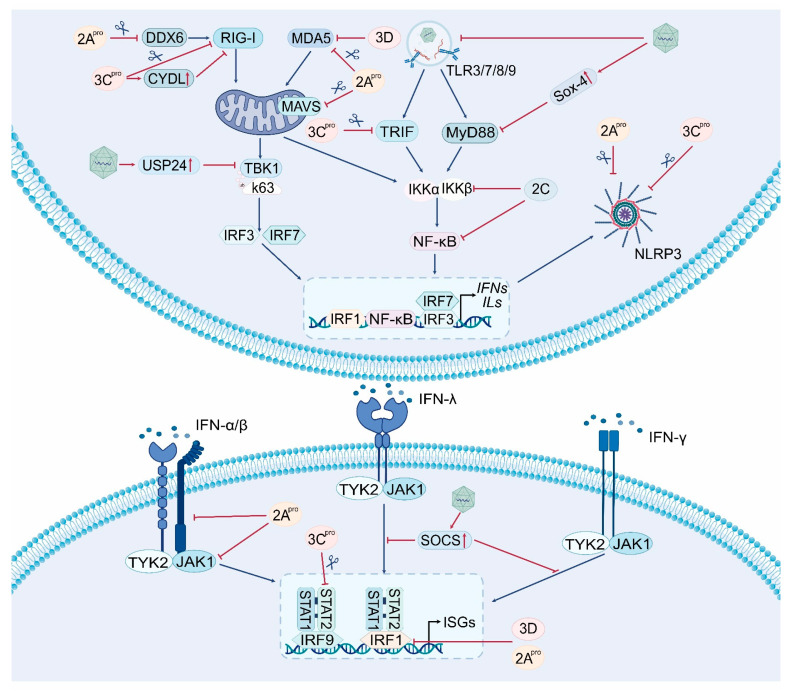
Innate immune evasion strategies of EV-A71. EV-A71 infection upregulates USP24, Sox4, and SOCS, which downregulate TBK1 polyubiquitination, inhibit Myd88 activity, and suppress JAK-STAT signaling. The virus also disrupts endosomal structures associated with TLR3/7/8/9. The 2A^pro^ hydrolyzes key factors involved in innate immune signaling, including MDA5, MAVS, NLRP3, and IFNAR1, blocking signal transduction. The 2C protein targets IKKα and IKKβ, preventing their phosphorylation, reducing p65/p50 heterodimer formation, and inhibiting NF-κB activation. The 3C^pro^ hydrolyzes RIG-I, NLRP3, TRIF, IRF7, and ISG3, and it upregulates CYLD to inhibit RIG-I ubiquitination. The 3D^pol^ inhibits MDA5 activity and downregulates IRF1 expression to inhibit IFN-γ signaling. Abbreviations: DDX6: DEAD-box helicase 6; CYLD: CYLD lysine 63 deubiquitinase; USP24: ubiquitin-specific protease 24; Sox4: sex-determining region Y-box 4; SOCS: suppressor of cytokine signaling; TBK1: TANK-binding kinase 1; IKKα/β: inhibitor of kappa B kinase α/β.

## Data Availability

Not applicable.

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
