# Peer review of "Recent Progress in Innate Immune Responses to Enterovirus A71 and Viral Evasion Strategies"

_ijms, 2024, doi:10.3390/ijms25115688_

Round 1
Reviewer 1 Report
Comments and Suggestions for Authors
Authors of the manuscript ”Recent Progresses in Innate Immunity and Immune Evasion by 2 Enterovirus A71” have the goal of providing a progress report of the Enterovirus A71 manipulation of the innate immunity to allow its propagation. Although, interesting, clarity in numerous places in the manuscript is needed. In addition, English needs improvement as there are abundant places that have run on and awkward sentences (see below). It is advised that authors have a native English speaker review this manuscript, as it is difficult to fully appreciate the impact as there are too many issues.
Major concerns are the following:
Spacing:
Correspondence: luoy@cqu.edu.cn; huifeng2022@cqu.edu.cn
Abstract
*Reword as there are three ”and” in the one sentence: Enterovirus 71 (EV-A71) is the major pathogen causing hand, foot, and mouth disease 8 (HFMD) in children worldwide, and it can lead to severe gastrointestinal, pulmonary, and neuro-9 logical complications.
*Reword abilities to ability:
Thus, the virus-host interactions and conflicts, as well 13 as the abilities to govern biological events at this first line of defense contribute significantly to the 14 pathogenesis and outcome of EV-A71 infection.
*Awkward sentence, please reword
In this review, we update recent studies about the signal transduction pathways in the course of host innate immunity to EV-A71 infection.
*Two “and” in this sentence; please reword
A better understanding of the interplay between EV-A71 and the host innate immunity will cer-18 tainly contribute to better comprehend disease progression and may unravel potential antiviral tar-19 gets and strategies that can improve patient outcomes.
Introduction
*Reword as there are three ”and” in the one sentence:
It was initially isolated from a child with disease of the 26 central nervous system in California in 1969 [1], and is a major pathogen causing hand, 27 foot and mouth disease (HFMD) that primarily affects infants and young children under 28 the age of five [2].
*Remove causing polio from this sentence.
(the most well-known and best-29 studied enterovirus causing polio)
*Awkward sentence, reword
The pathogenesis of EV-A71 infection, especially its mechanisms inducing severe 42 neurological complications,………
*Citation is needed
there is also growing appre-49 ciation of the impacts of effector-triggered immunity (ETI) that senses the activity of path-50 ogen-encoded effectors.
*Authors are encouraged to elaborate on the following:
it is well known that EV-A71 has developed sophisti-51 cated strategies to subvert antiviral innate immunity.
*The review should be organized as the authors stated in the following and to really to organize it with “sophisticated strategies to subvert anti-viral innate immunity”
Here we review recent progress in 52 understanding how EV-A71 infections trigger host innate immune responses. We also dis-53 cuss how EV-A71 targets the innate immune signaling pathways as a counteraction to 54 favor viral replication and propagation.
*It is confusing to have section 2. Past epidemiology of EV-A71-associated HFMD and 3. The molecular virology and life cycle of EV-A71 when they don’t round to the review structure and aims.
*Awkward sentence, reward
were all documented for the epidemiology of EV-A71 by 1990 [12-16].
Poor English
* Another epidemics happened
Inconsistent labelling of the virus. Please check whole manuscript
* EV71
*What are the actual cases and deaths in the following countries:
More recently, sporadic cases of EV71 infections have 72 also been reported in European countries including Denmark, France, Germany, Spain 73 and Poland [23-27].
*Incorrect word. Must use pandemic
In sum, the past decades have witnessed a significant increase in epi-74 demic activity of EV-A71 throughout the world.
*Spacing needs improvement
pseudo-T = 3
*figure 1 is pixelated. Having one sentence in the manuscript to describe the figure is not doing justice to Figure 1. Please cite this figure more and describe efficiently to value the true potential of this figure.
*Run on sentence, please reword
After enter-91 ing the host cell, the virus releases its genome into the cytoplasm through a pore on the 92 endosomal membrane [32-34], thereby directing the polyprotein translation as well as co-93 translationally proteolytic cleavage process that generates all mature viral proteins and 94 functional cleavage intermediates required for further replication.
*Punctuation is difficult in that it makes the sentence difficult to understand if pausing at the ,
and like 96 all positive-strand RNA viruses, in a non-conservative manner (i.e. new positive strands 97 outnumbering new negative strand RNA).
*Punctuation is difficult in that it makes the sentence difficult to understand if pausing at the ,
The translated polypeptides are then cleaved into 11 major proteins, including VP1-4 for viral 108 capsid assembly, and 2A-C, 3A-D for replication of viral genetic material.
*Same comment to Figure 2 as I put for figure 1 (*figure 2 is pixelated. Having one sentence in the manuscript to describe the figure is not doing justice to Figure 2. Please cite this figure more and describe efficiently to value the true potential of this figure.) In addition, all acronyms should be spelled out too.
*Repeat of sentence
As for EV-A71, understanding of the host innate immunity has been greatly improved in 134 the last decade.
*Please clarify in the whole manuscript. It is not clear what the authors are trying say and it is confusing when the authors are not focused to what they are talking about. For example, all under Innate immune evasion strategies of EV-A71 section:
-sophisticated strategies to subvert….. the antiviral innate response is different than subvert the innate response
And this is different than conteracting the host innate immunity to promote viral propagation.
-focus on the diverse strategies that EV-A71 applies 203 to counteract host innate immunity to promote viral propagation
*Same comment to Figure 3 as I put for figure 1 and 2 (*figure 3 is pixelated. Having one sentence in the manuscript to describe the figure is not doing justice to Figure 3. Please cite this figure more and describe efficiently to value the true potential of this figure.) In addition, all acronyms should be spelled out too.
*Spacing issues
Yet, as mentioned above,EV-A71
*Evade and interfering are not synonyms, so the heading should be reconsidered.
Innate immune evasion strategies of EV-A71 section
*Comment: One can seen the different level of academic writing that is used by the authors from section 5 compared to the rest of the paper. This discourse makes it difficult to appreciate the full potential of the manuscript.
*Comment: It might be clearer and more meaningful if authors visualize in a figure or table EV-A71’s strategies to manipulate the innate immune system and direct the attention to either: 1. Evasion strategy or 2, subversion strategy or 3 counteracting strategy to promote viral propagation.
For example, what does 5.1 manipulating PRRs in regards to the strategies above? Etc……
The table should be explicit to and cited back to the descriptive text in the manuscript.’
*Spacing issues
cleavage[108].
*Awkward sentence, reword
The availability of EV-A71 vaccines in 355 China but not any other country thus greatly enhances the chances that they may have 356 reduced efficacy against emerging new strains overtime
*Awkward sentence, reword
As such, effort is required 357 urgently to uncover the changes in the virus-host relationship that allow it to emerge as a 358 cause of EV-A71-associated severe neurological complications.
*Awkward sentence, reward
Only in recent years, however, have we begun to understand how EV-A71 in- 361 duces as well as antagonizes host innate immunity, and how this is associated with viral 362 pathogenesis.
*Poor English as these are not questions nor full sentences, reward
Whether this is because PRR response can favor production of type III 366 over type I IFNs, or whether EV-A71 evasion strategies themselves act in a way which 367 dampens type I IFN response more robustly?
And
Whether EV-A71 also counteracts activation 368 and signaling pathways of ETI initiated by viral protease activity?
And
In addition to the innate 369 immune signal cascades mentioned above, whether and how does other key components 370 of cellular innate immunity contribute to host protection against EV-A71 infection?
*Spacing
review; J.W.
*Poor English
provided intelectural inputs;
Comments on the Quality of English Language
Needs a lot of work in most places except section 5.
Author Response
Authors of the manuscript “Recent Progresses in Innate Immunity and Immune Evasion by 2 Enterovirus A71” have the goal of providing a progress report of the Enterovirus A71 manipulation of the innate immunity to allow its propagation. Although, interesting, clarity in numerous places in the manuscript is needed. In addition, English needs improvement as there are abundant places that have run on and awkward sentences (see below). It is advised that authors have a native English speaker review this manuscript, as it is difficult to fully appreciate the impact as there are too many issues.
We deeply value the reviewer's dedication in critiquing this manuscript. While we were somewhat taken aback by the reviewer's emphasis on language rather than scientific content, we found these comments to be immensely constructive. In response, we have addressed these points as detailed below, and we are confident that the manuscript has been enhanced as a result. Our point-by-point response to the reviewer’ comments appears below, with the reviewers’ comments in red/italic font.
Major concerns are the following:
Abstract
*Reword as there are three ”and” in the one sentence: Enterovirus 71 (EV-A71) is the major pathogen causing hand, foot, and mouth disease 8 (HFMD) in children worldwide, and it can lead to severe gastrointestinal, pulmonary, and neuro-9 logical complications.
Thanks for addressing the specific issue. As demonstrated in the examples below, the use of "and" three times is common in scientific writing. The key is to ensure that the sentence flows well and aligns with the context. However, we agree that splitting the sentence after the second "and" would enhance readability. We have made this adjustment in various sections of the manuscript.
Below shows a few examples of three “and” expression in publications:
Paulina Cruz de Casas et al., Nature Reviews Immunology. 2024, 24(5):358-374.
(1) We will discuss how lymph node heterogeneity impacts on cellular and humoral immune responses and the implications for vaccination, tumour development and tumour control by immunotherapy.
(2) In this Review, we highlight the cell types and molecular factors that contribute to LN heterogeneity during homoeostatic conditions and cancer and explain how they shape the nature of the immune responses generated at these sites.
(3) To this end, LNs need to compartmentalize lymphocytes in B cell and T cell areas but must also promote the interaction of these lymphocytes and other immune cells to orchestrate immune responses in space and time.
Carola G. Vinuesa et al., Science. 2023, 380(6644):478-484.
(4) These variants occur in TLR7 and in molecules involved in IFN I production and signaling, including OAS1, TYK2, andIFNAR2.
Liu Siqi et al., Cell. 2023, 186(10):2127-2143.e22.
(5) Mechanistically, Il24 expression depends upon both epithelial IL24-receptor/STAT3 signaling and hypoxia-stabilized HIF1α, which converge following injury to trigger autocrine and paracrine signaling involving IL-24-mediated receptor signaling and metabolic regulation.
*Reword abilities to ability:
Thus, the virus-host interactions and conflicts, as well 13 as the abilities to govern biological events at this first line of defense contribute significantly to the 14 pathogenesis and outcome of EV-A71 infection.
We have corrected this.
*Awkward sentence, please reword
In this review, we update recent studies about the signal transduction pathways in the course of host innate immunity to EV-A71 infection.
We have simplified the sentence by rewording it to “In this review, we update recent progresses on host innate immune responses to EV-A71 infection”.
*Two “and” in this sentence; please reword
A better understanding of the interplay between EV-A71 and the host innate immunity will cer-18 tainly contribute to better comprehend disease progression and may unravel potential antiviral tar-19 gets and strategies that can improve patient outcomes.
As we mentioned above, we have rephased this sentence.
Introduction
*Reword as there are three ”and” in the one sentence:
It was initially isolated from a child with disease of the 26 central nervous system in California in 1969 [1], and is a major pathogen causing hand, 27 foot and mouth disease (HFMD) that primarily affects infants and young children under 28 the age of five [2].
We have split the sentence at the second “and”, making it shorter and easier to read through while it delivers exactly the same information.
*Remove causing polio from this sentence.
(the most well-known and best-29 studied enterovirus causing polio)
We have removed “causing polio” from the sentence as the reviewer suggested.
*Awkward sentence, reword
The pathogenesis of EV-A71 infection, especially its mechanisms inducing severe 42 neurological complications,………
We have rephased the sentence as the reviewer suggested.
*Citation is needed
there is also growing appre-49 ciation of the impacts of effector-triggered immunity (ETI) that senses the activity of path-50 ogen-encoded effectors.
We have added literatures at the end of the sentence as the reviewer suggested.
*Authors are encouraged to elaborate on the following:
it is well known that EV-A71 has developed sophisti-51 cated strategies to subvert antiviral innate immunity.
We thank the reviewer for ringing this bell. Here in “Introduction” we aimed to help the readers catching the scope as well as the very brief scientific concept of our review. Nonetheless, we indeed have added a sentence to the text, i.e. “For instance, it encodes two viral proteases that interfere with host innate immune signaling, as described in detail below.” We have also switched the word “subvert” to “escape” to avoid any confusion.
*The review should be organized as the authors stated in the following and to really to organize it with “sophisticated strategies to subvert anti-viral innate immunity”
Here we review recent progress in 52 understanding how EV-A71 infections trigger host innate immune responses. We also dis-53 cuss how EV-A71 targets the innate immune signaling pathways as a counteraction to 54 favor viral replication and propagation.
*It is confusing to have section 2. Past epidemiology of EV-A71-associated HFMD and 3. The molecular virology and life cycle of EV-A71 when they don’t round to the review structure and aims.
We thank the reviewer for pointing this out. As we outlined in the manuscript, the focus is the interactions between EV-A71 infection and host innate immune responses. It mainly includes: (1) how host innate immune system responses to virus infection? (i.e. Section 4); (2) how the virus evades host innate immune surveillance to favor virus propagation? (i.e. Section 5). We believe these contents are well documented in the title, abstract, and the main text of the manuscript. We also believe that the manuscript would have lacked fundamental background if “Past Epidemiology of EV-A71-Associated HFMD” and “The Molecular Virology and Life Cycle of EV-A71” (Sections 2 and 3) was not presented at first. Nonetheless, we agree that we could have organized these sections better (or at least less misleading). Therefore, we have rephrased these two sentences accordingly.
*Awkward sentence, reward
were all documented for the epidemiology of EV-A71 by 1990 [12-16].
We have rephased the expression as the reviewer suggested.
Poor English
* Another epidemics happened
We have rephased the expression as the reviewer suggested.
Inconsistent labelling of the virus. Please check whole manuscript
* EV71
We apologize for this inconsistence. We have corrected this and checked the entire manuscript to avoid this mistake.
*What are the actual cases and deaths in the following countries:
More recently, sporadic cases of EV71 infections have 72 also been reported in European countries including Denmark, France, Germany, Spain 73 and Poland [23-27].
We thank the review for bringing this up, and we have gone through these publications for details. However, the case numbers in all these studies only represented the confirmative cases in certain regions that fit the scope of each study. For example, only six polish patients with neurological symptoms were included in the research carried out in Poland. Nonetheless, we have added specific information in Denmark and France to the text as suggested, so the readers can sense the sporadic outbreaks of EV-A71 infection in different regions.
*Incorrect word. Must use pandemic
In sum, the past decades have witnessed a significant increase in epi-74 demic activity of EV-A71 throughout the world.
We have corrected this as suggested.
*Spacing needs improvement
pseudo-T = 3
“pseudo-T = number” has been used by virologists to describe the capsid structure of virions (please see below examples (4) and (5)). More often, they use the expression of “pseudo T = number” in their publications (please see example (1) to (3) shown in below). We now switch to this latter form in the manuscript. Nonetheless, either “pseudo-T = number” or “pseudo T = number” is well-accepted by virologists.
Below shows a few examples of “pseudo T = number” expression in publications:
- Together they form the icosahedral shell with a pseudo T = 3 arrangement that encapsidates the viral genome. (Baggen J et al., Nature Reviews Microbiology. 2018, 16, pages368–381)
- The conventional penta-symmetron formed by the capsomeres is replaced by a large vertex complex in the pseudo T = 25 (Reddy HK et al., Elife. 2019, 8:e48496.)
- The strong similarity between FLiP and another member of the structural lineage, bacteriophage PM2, extends to the capsid organization (pseudo T = 21 dextro). (Laanto E et al., PNAS. 2017, 114(31):8378-8383.)
- which form small icosahedral, non-enveloped particles with a pseudo-T = 3 symmetry (Olivier Le Gall et al., Arch Virol. 2008,153(4):715-27.
- They have a pseudo-T = 25 triangulation number with at least 12 different proteins composing the virion. (José Gallardo et al., Int J Mol Sci. 2021, 22(10):5240.)
*figure 1 is pixelated. Having one sentence in the manuscript to describe the figure is not doing justice to Figure 1. Please cite this figure more and describe efficiently to value the true potential of this figure.
We apologize for the pixelated figure 1, and we agree that citing it in just one sentence is not doing justice to the figure. We have (1) reorganized figure 1 and the figure legend by adding several sequentially key steps of the virus life cycle; (2) cited figure 1 when necessary.
*Run on sentence, please reword
After enter-91 ing the host cell, the virus releases its genome into the cytoplasm through a pore on the 92 endosomal membrane [32-34], thereby directing the polyprotein translation as well as co-93 translationally proteolytic cleavage process that generates all mature viral proteins and 94 functional cleavage intermediates required for further replication.
We have split the sentence into two shorter ones and rephrased them as following in the revised manuscript: Once the virus enters a host cell, it releases its RNA genome into the cytoplasm through a pore on the endosomal membrane. This RNA genome is then directed to the polyprotein translation and co-translationally proteolytic cleavage process, generating all mature viral proteins and functional cleavage intermediates required for further replication.
*Punctuation is difficult in that it makes the sentence difficult to understand if pausing at the ,
and like 96 all positive-strand RNA viruses, in a non-conservative manner (i.e. new positive strands 97 outnumbering new negative strand RNA).
We have rephrased this entire sentence as following: Like all positive-strand RNA viruses, this occurs in close association with replication organelles (infection-induced membrane structures) and in a non-conservative manner (i.e. new positive strands outnumbering new negative strand RNA, Fig. 1â‘£).
*Punctuation is difficult in that it makes the sentence difficult to understand if pausing at the ,
The translated polypeptides are then cleaved into 11 major proteins, including VP1-4 for viral 108 capsid assembly, and 2A-C, 3A-D for replication of viral genetic material.
We have rephrased this sentence as following: The translated polypeptides are then cleaved into 11 major proteins, including VP1-4 for viral capsid assembly and 2A-3D for replication of viral RNA genome.
*Same comment to Figure 2 as I put for figure 1 (*figure 2 is pixelated. Having one sentence in the manuscript to describe the figure is not doing justice to Figure 2. Please cite this figure more and describe efficiently to value the true potential of this figure.) In addition, all acronyms should be spelled out too.
As we said earlier, we apologize for the pixelated figure. We have cited figure 2 when necessary. Moreover, we have checked the acronyms and spelled out all of them.
*Repeat of sentence
As for EV-A71, understanding of the host innate immunity has been greatly improved in 134 the last decade.
We have removed “understanding of the host innate immunity has been greatly improved in the last decade” and rephrased the sentences as following in the revised manuscript: Regarding EV-A71, it has been demonstrated that TLRs including TLR3, TLR7, and TLR9, as well as the RLR protein, melanoma differentiation-associated gene 5 (MDA5), all mediate IFN responses against the virus.
*Please clarify in the whole manuscript. It is not clear what the authors are trying say and it is confusing when the authors are not focused to what they are talking about. For example, all under Innate immune evasion strategies of EV-A71 section:
-sophisticated strategies to subvert….. the antiviral innate response is different than subvert the innate response
And this is different than conteracting the host innate immunity to promote viral propagation.
-focus on the diverse strategies that EV-A71 applies 203 to counteract host innate immunity to promote viral propagation
In this section, we aimed to discuss how EV-A71 evades host innate immune responses, and we indeed summarized various viral strategies escaping host innate immunity in this section. We think the paper flows well from (a) viral evasion strategies by manipulating PRRs, through (b) strategies to disrupt key adaptors of the PRR signaling pathways and (c) strategies to dampen transcription factor activation/function, to (d) strategies to antagonize IFN-initiated immune responses. Also as described in below, “evade/evasion” is widely used by microbiologists, virologists and immunologists to describe the reactions that pathogens escape from the surveillance of host immune responses. Researchers in the field of host immunity-virus interaction do not tend to classify them as “subvert” or “counteract”, even though both words have been frequently used in literatures. Nonetheless, we do agree that the paper should not cause any unnecessary confusion. Therefore, we have reconsidered the subtitles of section 5. Also, we have used the word “counteract” in one specific sentence (Equally remarkable, it was shown that EV-A71 can also counteract inflammasome activation through NLRP3 cleavage at the G493^L494 or Q225^G226 junction by virus 2Apro and 3Cpro), but not other places. We did this because this is the word the authors used in their publication (Wang et al., Cell Reports. 2015, 12(1):42-48.).
*Same comment to Figure 3 as I put for figure 1 and 2 (*figure 3 is pixelated. Having one sentence in the manuscript to describe the figure is not doing justice to Figure 3. Please cite this figure more and describe efficiently to value the true potential of this figure.) In addition, all acronyms should be spelled out too.
Again, we apologize for the pixelated figure, and we have cited figure 3 when necessary. Moreover, we have checked the acronyms and spelled out all of them.
*Spacing issues
Yet, as mentioned above,EV-A71
We have corrected this.
*Evade and interfering are not synonyms, so the heading should be reconsidered.
Innate immune evasion strategies of EV-A71 section
“Evade/evasion” is widely used by microbiologists, virologists and immunologists to describe the reactions that pathogens escape from the surveillance of host immune responses. However, we do agree with the reviewer that “interfering” is not a synonym of evade/evasion. We have thus reconsidered and changed the subtitles of section 5 so that they do not cause unnecessary confusion.
*Comment: One can seen the different level of academic writing that is used by the authors from section 5 compared to the rest of the paper. This discourse makes it difficult to appreciate the full potential of the manuscript.
We have checked the entire manuscript thoroughly, and asked native English speaker Dr. Ya-Wei Xiong at Centers for Disease Control and Prevention, United States reviewing this manuscript so that it can deliver the scientific value more efficiently.
*Comment: It might be clearer and more meaningful if authors visualize in a figure or table EV-A71’s strategies to manipulate the innate immune system and direct the attention to either: 1. Evasion strategy or 2, subversion strategy or 3 counteracting strategy to promote viral propagation.
For example, what does 5.1 manipulating PRRs in regards to the strategies above? Etc……
The table should be explicit to and cited back to the descriptive text in the manuscript.’
We thank the reviewer for this idea of adding a new table to the manuscript. However, we have to point out that evasion strategy is a well-accepted concept in the field of host immunity-virus interaction, and researchers do not tend to classify them as the reviewer suggested. Nonetheless, we do hope the manuscript tells its story in a better way. As the reviewer suggested, we have included a new table in the revised manuscript. In it, we specify the strategies, targets, and the underlying mechanisms so that the reader can get these details more easily.
*Spacing issues
cleavage[108].
We have corrected this.
*Awkward sentence, reword
The availability of EV-A71 vaccines in 355 China but not any other country thus greatly enhances the chances that they may have 356 reduced efficacy against emerging new strains overtime
Again, we thank the review for pointing this out. In order to make the context here more reasonable and much easier to read through. We have rephrased this sentence as: The availability of EV-A71 vaccines in China, but not any other country, increases the probability of pathogen evolution, leading to the emergence of new strains of EV-A71 over time [116].
*Awkward sentence, reword
As such, effort is required 357 urgently to uncover the changes in the virus-host relationship that allow it to emerge as a 358 cause of EV-A71-associated severe neurological complications.
We have rephrased this sentence as: As such, more research is urgently required to understand the complex molecular interplay between the host and the virus.
*Awkward sentence, reward
Only in recent years, however, have we begun to understand how EV-A71 in- 361 duces as well as antagonizes host innate immunity, and how this is associated with viral 362 pathogenesis.
We have rephrased this sentence as: However, we have only begun to understand how EV-A71 induces as well as antagonizes host innate immunity, and how this is associated with viral pathogenesis in recent years.
*Poor English as these are not questions nor full sentences, reward
Whether this is because PRR response can favor production of type III 366 over type I IFNs, or whether EV-A71 evasion strategies themselves act in a way which 367 dampens type I IFN response more robustly?
And
Whether EV-A71 also counteracts activation 368 and signaling pathways of ETI initiated by viral protease activity?
And
In addition to the innate 369 immune signal cascades mentioned above, whether and how does other key components 370 of cellular innate immunity contribute to host protection against EV-A71 infection?
Following the suggestion of the reviewer, we have rephrased the future directions as below:
For instance, what makes EV-A71 highly neurotrophic? What is the molecular basis behind the phenomenon that EV-A71 infection preferentially triggers type III IFN responses? Whether EV-A71 dampens type I IFN response more robustly? Whether EV-A71 can also evade viral protease activity-dependent ETI activation and signaling? In addition to the innate immune signal cascades mentioned above, whether and how do other key components of cellular innate immunity contribute to host protection against EV-A71 infection?
*Spacing
review; J.W.
We have double checked the space in the manuscript we submitted earlier, and found that no space was missing at this specific site.
*Poor English
provided intelectural inputs;
We apologize for this typo. We have corrected this in the revised manuscript.

Reviewer 2 Report
Comments and Suggestions for Authors
I read the paper with great interest. Sincerely is not easy to do suggestion due to high quality of paper. Figure are excelent
only few minor minor reccomendations:
1. About paragraph 4: explain better the role of PAMPs
2. Conclusion: give some proposal that came from your deep and well done review
3. Add methods section
Author Response
We greatly appreciate the time spent by the reviewer to critique this manuscript. These comments were very helpful. We have responded to these comments as outlined below, and believe the manuscript is improved as a result.
Our point-by-point response to the reviewer’ comments appears below, with the reviewers’ comments in red/italic font.
Comments and Suggestions for Authors
I read the paper with great interest. Sincerely is not easy to do suggestion due to high quality of paper. Figure are excelent
only few minor minor reccomendations:
- About paragraph 4: explain better the role of PAMPs
We agree that the role of PAMPs in triggering host innate immune responses is fundamental to understand the innate immune signaling pathways, and we thank the reviewer for pointing this out. We have (1) added a few words in the 2nd paragragh of the “Innate immune responses to enterovirus infections” section, specifying that the PAMPs include the viral RNA genome released in host cells and the double-stranded RNA (dsRNA) generated during viral replication; (2) added this information in Fig.2 in order that the figure correlates well with the text, and that the readers can readily get to the point from both sides.
- Conclusion: give some proposal that came from your deep and well done review
We would like to deeply thank the review for this very positive comment on our manuscript. We agree that a few more words in the end will help to wrap up the entire story. Since we have proposed several interesting questions/directions in the last paragragh of the manuscript we submitted earlier, here in this version we have (1) added a new puzzling question which we believe is important to the field; (2) added a very brief concluding paragragh to sum up our current opinions.
- Add methods section
We understand that the methods section is usually fundamental for systematic reviews and Meta analysis. Nonetheless, we have not seen this section in any of other types of review papers. We would be more than happy to do so if we plan to draft a systematic review and Meta analysis in the near future.

Round 2
Reviewer 1 Report
Comments and Suggestions for Authors
The second read of the manuscript titled “Recent Progress in Innate Immune Responses to Enterovirus 2 A71 and Viral Evasion Strategies” provides a better flow of the author’s goals for this review. In response to the authors’ comments and examples of having two or three “ands” in a sentence, this just leads to run on sentences and lack of understanding of the sentence. The manuscript should not be hard to understand for the reader. Having said this, almost all the points raised before were addressed but there are still minor points to consider in the next iteration of the manuscript. In addition, Table 1 provides a clear message from the authors and allows the reader to follow the authors’ goals for the manuscript.
Minor points.
*“In addition, we discuss the underlying 19 strategies employed by EV-A71 to escape this intracellular response.”
What is meant by “this intracellular response” as it is too vague here.
*The use of “For example” is eight times in the manuscript. One can use the thesaurus to add academic nature to these sentences.
“although textbooks consider enteroviruses”
This is more colloquial wording (in bold) and would be better if it is more academic
*Figure 1 does not have a title like Figure 2 and 3 does.
*spacing issues
pores(â‘ -â‘¡).
3Cpro cleavage[109].
*Awkward sentence
“In comparison, although it has been shown that MyD88 is targeted by EV-A71 as 299 well, the mechanisms vary dramatically.”
In comparison to what? What mechanisms?
*No correct sentence structures in the following.
“Whether EV-A71 dampens type I IFN response more robustly?”
“Whether EV-A71 can also evade viral protease activity-dependent ETI activation and signaling?”
Comments on the Quality of English LanguageJust a few minor issues that can easily be corrected. See comments above
Author Response
The second read of the manuscript titled “Recent Progress in Innate Immune Responses to Enterovirus 2 A71 and Viral Evasion Strategies” provides a better flow of the author’s goals for this review. In response to the authors’ comments and examples of having two or three “ands” in a sentence, this just leads to run on sentences and lack of understanding of the sentence. The manuscript should not be hard to understand for the reader. Having said this, almost all the points raised before were addressed but there are still minor points to consider in the next iteration of the manuscript. In addition, Table 1 provides a clear message from the authors and allows the reader to follow the authors’ goals for the manuscript.
We thank the reviewer for taking out of his/her time to re-evaluate the manuscript. These suggestions have greatly contributed to improving the academic quality of our article. We have responded to these suggestions as outlined below and believe that the manuscript has been improved as a result. Our point-by-point response to the reviewers' comments is shown below, with the reviewers' opinions highlighted in italic font.
Minor points.
*“In addition, we discuss the underlying 19 strategies employed by EV-A71 to escape this intracellular response.”
What is meant by “this intracellular response” as it is too vague here.
We have rewritten this sentence “In addition, we discuss the underlying strategies employed by EV-A71 to escape host innate immune response.” to express it clearly.
*The use of “For example” is eight times in the manuscript. One can use the thesaurus to add academic nature to these sentences.
We thank the reviewer for taking a close look at this manuscript. We have checked the main text. With regard to the 9 “For example” expression, we have: (1) replaced two of the “For example” expression with “For instance" (line 76-77, line 249-250), (2) rephrased two of the "For example” expression as “Among them” (line 158-159) and “In agreement” (line 323) according to the context, (3) deleted a “for example” (line 315).
“although textbooks consider enteroviruses”
This is more colloquial wording (in bold) and would be better if it is more academic
We have rephrased the sentence as “although enteroviruses have been recognized as non-enveloped lytic viruses released from ruptured host cells”.
*Figure 1 does not have a title like Figure 2 and 3 does.
Although we are surprised this can be a problem, we have changed the previous title of Figure 1 to “Illustration of EV-A71 life cycle”.
*spacing issues
pores(â‘ -â‘¡).
3Cpro cleavage[109].
We have corrected these spacing issues.
*Awkward sentence
“In comparison, although it has been shown that MyD88 is targeted by EV-A71 as 299 well, the mechanisms vary dramatically.”
In comparison to what? What mechanisms?
In order to make the sentence clear to the readers, we have rewritten this sentence as “In comparison to adaptor proteins being cleaved directly, MyD88 is targeted by EV-A71 as well, yet it occurs in an indirect manner”.
*No correct sentence structures in the following.
“Whether EV-A71 dampens type I IFN response more robustly?”
“Whether EV-A71 can also evade viral protease activity-dependent ETI activation and signaling?”
We have rephrased these sentences.